# A Retrospective Study on Patient-Specific Predictors for Non-Response to Sacroiliac Joint Injections

**DOI:** 10.3390/ijerph192315519

**Published:** 2022-11-23

**Authors:** Rahul S. Chandrupatla, Bahar Shahidi, Kelly Bruno, Jeffrey L. Chen

**Affiliations:** 1School of Medicine, University of California San Diego, 9500 Gilman Drive, La Jolla, CA 92093, USA; 2Department of Orthopaedic Surgery, University of California San Diego, 9500 Gilman Drive (MC0863), La Jolla, San Diego, CA 92093, USA; 3Department of Anesthesiology, Division of Pain Medicine, University of California San Diego, 9300 Campus Point Drive (MC 7651), La Jolla, CA 92037, USA

**Keywords:** sacroiliac joint, low back pain, corticosteroid, chronic pain

## Abstract

Intra-articular or peri-articular corticosteroid injections are often used for treatment of sacroiliac joint (SIJ) pain. However, response to these injections is variable and many patients require multiple injections for sustained benefit. In this study, we aim to identify patient-specific predictors of response or non-response to SIJ injections. Identification of these predictors would allow providers to better determine what treatment would be appropriate for a patient with SIJ pain. A retrospective review of 100 consecutive patient charts spanning a 2-year period at an academic multi-specialty pain center was conducted and a multivariate regression analysis was used to identify patient-specific predictors of response to SIJ injections. Our analysis identified that a history of depression and anxiety (OR: 0.233, 95%CI: 0.057–0.954) and increased age (OR: 0.946, 95%CI: 0.910–0.984) significantly reduced the odds of responding to injections. We also found that the associated NPRS score change for SIJ injection responders was less than the minimally clinically significant value of a 2-point differential, suggesting that reported changes in pain scores may not accurately represent a patient’s perception of success after SIJ injection. These findings warrant further investigation through a prospective study and can potentially influence clinical decision making and prognosis for patients receiving SIJ injections.

## 1. Introduction

SIJ pain is a common cause of chronic low back pain (CLBP) as an estimated 15–21% of all CLBP is due to dysfunction of the SIJ [1]. The etiology of SIJ pain is usually due to a combination of axial loading and rotation. Common causes of SIJ pain include arthritis, spondyloarthropathies, ligamentous and muscular injuries, and enthesopathy [1].

The diagnosis of SIJ pain remains challenging as the characteristics of the pain can vary between patients. Pain referral patterns are very common and variable in SIJ pain due to the size and heterogeneity of the joint [1]. The reference standard for diagnosing SIJ pain has been to use a low volume anesthetic block, and multiple studies have used the threshold of an improvement of 75% or greater in pain while performing previously painful movements to establish SIJ dysfunction [2].

Initial treatment of SIJ pain includes ice and physical therapy [1]. Patients who do not show improvement with these measures often undergo intra-articular or peri-articular SIJ corticosteroid injections. However, even with the use of SIJ injections, the response is variable and multiple injections are often performed in an attempt to manage pain [3]. While mechanical or inflammatory components have been hypothesized to account for differences in why certain patients respond better to SIJ injections than others, there is an overall lack of research in predictors of responsiveness to SIJ injections. These predictors could improve long-term clinical outcomes and lower costs of care by helping to guide decisions on a patient’s care. For instance, patients that are identified to be less likely to respond to SIJ injections can be evaluated for different treatments such as regenerative therapies (bone marrow aspirate concentrate (BMAC), platelet-rich plasma (PRP) injections and prolotherapy) or sacroiliac joint fusion [1] earlier.

Previous studies have investigated pain referral patterns and SIJ pain with varying results. The area 1cm inferomedial to the posterior inferior iliac spine (PSIS), which is commonly demonstrated to be painful in the “Fortin finger test”, was described to be specifically present in SIJ pain [4]. Patient specific predictors such as demographic and clinical variables including previous medical, psychiatric, and surgical history are easily accessible to the clinician and may help to determine if a patient would benefit from a SIJ corticosteroid injection before it is even performed. Therefore, investigation of how these patient-specific factors can affect response to SIJ injections is warranted. This study aims to investigate the relationship between 10 patient specific demographic and clinical variables and determine effectiveness for predicting response to a sacroiliac joint injection.

## 2. Materials and Methods

This was a retrospective chart review of 100 consecutive patients at an academic multi-specialty pain center who received one or more SIJ injections under fluoroscopic guidance between 2015 and 2017. Patients were referred through their primary care physicians and had chronic (>3 months duration) pain. Inclusion criteria for the study were: age 18 years or older, had undergone sacroiliac joint injection(s) with a pre-procedure and post-procedure clinic visit documented in their medical records, and at least one of the six positive provocative SIJ exam tests described by Laslett et al. (distraction, thigh thrust, Gaenslen, compression, and sacral thrust) [5] recorded prior to injection. Exclusion criteria included incomplete documentation of pre-procedure and/or post-procedure and additional interventions to the lower back/SIJ area (lumbar epidural, sacroiliac fusion, etc) before first post-SIJ injection clinic visit. SIJ injections involved the insertion of a 22- or 25-gauge 3.5-to-5-inch spinal needle into the sacroiliac joint 1–2 cm cephalad of the inferior posterior margin of the SIJ, or the superior pelvic brim of the arcuate line. This position was confirmed via AP and/or lateral fluoroscopy under contrast using Omnipaque 240. Then, a solution containing a combination of local anesthetic (bupivacaine 0.25%) and corticosteroid (triamcinolone, methylprednisolone, dexamethasone, or betamethasone) was injected into the joint space. This study was approved by the local institutional review board.

### 2.1. Measurements

The primary outcome of this study was whether a patient responded to their first SIJ injection. A patient was classified as a “responder” if they self-reported a ≥50% improvement on a visual analogue scale after SIJ injection that lasted ≥2 weeks to distinguish between corticosteroid and local anesthetic. The 50% improvement threshold was based on prior literature identifying meaningful clinical improvement for SIJ injection [6]. Secondary outcome measures included change in Numeric Pain Rating Scale (NPRS) score between the pre-injection clinic visit and at first follow-up visit, as well as duration of improvement as subjectively reported by patients in subsequent clinic visits. The first follow-up visit was scheduled for 6 weeks post-injection. The NPRS has been validated in patients with low back pain [6]. Occupation, including working status, was obtained. Patient-specific predictors included demographic variables (age, gender, BMI, Opioid Risk Tool (ORT) score), pain referral pattern (lower back/buttocks only (A), lower back/buttocks with radiation down to knees (B), lower back/buttocks with radiation past knees (C), and lower back/buttocks + upper body pain (D)) (Figure 1), smoking status (nonsmoker, past smoker, and current smoker), and history of depression, anxiety, diabetes, hypertension, and lumbar spine surgery.

Pain referral patterns have been studied previously in SIJ populations [4]. The Opioid Risk Tool (ORT) is a questionnaire on prior drug use history and personal and family history of psychosocial problems. Hospital systems widely use the Opioid Risk Tool (ORT) to assist in the prediction of aberrant behaviors related to opioids [8], given that continued opioid use has been noted to lessen improvement in pain and disability in minimally invasive surgical management for SIJ pain [9]. In conjunction with a history of depression and/or anxiety, the ORT can gauge the patient’s psychiatric history, as patients with higher baseline levels of negative affect, including anxiety and depression, have been shown to have worse outcomes of pain severity and disability [10]. Other covariates such as history of spinal surgery [11], smoking [12], diabetes [13], and hypertension [14] were also included based on prior investigations demonstrating their influence on clinical outcomes in low back pain.

### 2.2. Statistical Analysis

A multivariate logistic regression model to determine significant predictors for responsiveness to SIJ injections was performed. This statistical analysis was performed using SPSS Version 26 (IBM Corp. Released 2019. IBM SPSS Statistics for Windows, Version 26.0. Armonk, NY, USA). This was a complete-case analysis. Model performance was assessed through the Nagelkerke R square of the logistic regression analysis. Two tailed independent student *t*-tests were used to compare primary and secondary outcome values for responders vs. non-responders listed in Table 1. *p* ≤ 0.05 was considered statistically significant. The sample size of N = 100 was determined based on the suggested criteria of 10 participants per independent variable of interest for multivariate regression models [15].

## 3. Results

In total, 140 patient charts were reviewed and 40 patient charts were excluded, leaving 100 charts for data analysis. Of the 100 patient charts reviewed, 40 (40%) were identified as responders and 60(60%) as non-responders. Average ± SD time to first follow-up visit was 2.6 ± 2.4 months. Table 1 lists the patient demographic characteristics for each group. Occupations were organized into 6 categories according to level of physical activity: “Manual Labor” (e.g., carpenter, professional athlete, nurse), “Sedentary Job” (e.g., office manager, student, writer), “Homemaker”, “Unemployed/Disability”, “Retired”, and “Unknown”. Responders demonstrated a mean ± SD improvement of 74 ± 16% after initial injection, a reduction in NPRS of 1.1 ± 2.4 points, and a duration of improvement of 4.1 ± 3.0 months. Non-responders demonstrated an improvement of 24 ± 28%, a reduction in NPRS of 0.25 ± 1.5 points, and a duration of improvement of 1.1 ± 1.3 months. The differences of mean improvement and duration of improvement between both groups were statistically significant (*p* < 0.001). The change in NPRS between both groups was statistically significant as well (*p* = 0.034). The difference of mean age in a univariate analysis was statistically significant (*p* = 0.019); however, there was no statistical difference between the BMI of both groups in the univariate analysis.

The mean ± SD age of patients was 57 ± 14 for responders vs. 64 ± 16 for non-responders (Figure 2A). There was a greater proportion of responders who were female compared to non-responders. A greater proportion of non-responders had spinal surgery compared to responders. Non-responders and responders had similar frequencies of smoking status. The most frequent pain referral pattern for non-responders was B (pain in buttocks radiating down one leg until knee) vs. A for responders (lower back/buttocks only) (Figure 1). Non-responders had a greater proportion of both depression and anxiety than responders (Figure 2B). A smaller proportion of non-responders had hypertension vs. responders. The percentage of patients with a history of diabetes was similar for both responders vs. non-responders.

The overall multivariate logistical regression explained 24.6% of the variance in the outcome of responsiveness (R^2^ = 0.246). The model revealed that a history of depression and anxiety and increased age significantly reduced the odds of responding to injections. In particular, patients with a history of depression and anxiety concomitantly were 76.7% less likely to be responders, and each additional year of age increased the odds of failure by 5.4%. Gender, history of lumbar spine surgery, smoking status, pain referral pattern, Opioid Risk Tool scores, history of hypertension, and history of diabetes were not predictive of response to injection (*p* > 0.144). (Table 2 and Table 3)

## 4. Discussion

This retrospective study aimed to identify patient-specific predictors of responsiveness to intra-articular SIJ corticosteroid injections. A history of depression and anxiety and increased age significantly reduced the odds of responding to injections.

Previous studies have primarily focused on physical exam findings and pain location as predictors for SIJ pain, with varying results [4,5,7]. Many of the patient-specific predictors in this study have not been studied directly in the context of only SIJ pain and treatment. A recent study by Cohen et al. [16] studying the effects of demographic variables on response to either epidural steroid injections, SIJ injections, and facet interventions for axial LBP revealed lower baseline pain score, depressive symptomatology, and obesity were associated with smaller pain outcomes. Our study reflects the effects of patient-specific predictors for SIJ injections alone, which may account for the differences in our results as well as the similarities with regard to depressive symptoms.

Age was associated with a decreased likelihood of responding to SIJ injections, which is consistent with other studies investigating SIJ and/or chronic low back pain and interventions [17,18]. Increased age has been associated with an increased probability of sacroiliac joint pain being the source of low back pain [18]. One explanation that has been offered is that elderly patients more often have progressive arthritic SIJ pain when compared to younger patients, who tend to have sources of pain that are self-limiting [17]. This arthritic SIJ pain is likely less responsive to corticosteroid injections and requires multiple injections for the same effect. Additionally, other studies have identified an association between increased comorbid disease burden and severity of self-reported pain [19], and elderly patients, on average, have a greater number of comorbidities than younger patients [20].

A history of depression and anxiety was associated with a reduced odds of responding to SIJ injections. This is consistent with previous studies suggesting psychosocial factors contribute to persistent pain in the spine [21]. Interestingly, a history of depression only or anxiety only was not associated with a significant change in responsiveness to SIJ injections. This suggests that a high psychological burden is necessary for patients’ response to be affected.

One unexpected finding was that despite responders subjectively reporting an improvement of greater than 50% pain reduction, the associated change in NPRS scores was not clinically significant (<2 point change) [6]. Importantly, this suggests that change in NPRS score may not accurately represent patient perception of success after SIJ injections. NPRS has been used in other studies to determine response to SIJ injections [22] and is often used as part of a clinician’s decision-making process for additional injections. If patients who experience >50% improvement in SIJ pain after injection do not have an NPRS pain score improvement by 2 points, then the SIJ injection could falsely be seen as a failure in this study. One potential reason for the discrepancy seen here is that most of the patients who get SIJ injections are seen in chronic pain clinics and are unlikely to have a singular chronic pain diagnosis. A singular NPRS score does not allow us to distinguish between various pain complaints, and so it is likely that patients’ NPRS scores reflect the status of all of their chronic pain versus specific SIJ pain only. This suggests the need in clinical charting to use metrics that can differentiate between multiple sources of pain for a patient. Previous studies have used multiple metrics other than NPRS to evaluate a patient’s response to SIJ injection, and some of these metrics, such as the Oswestry Disability Index for determining a patient’s functional status, may be helpful for clinical decision making after an initial SIJ injection [23]. Additionally, given the finding that non-responders were statistically significantly predicted by anxiety and depression, subjective improvement after SIJ injections could also be influenced by psychosocial factors, which need to be accounted for when determining response to treatment. This discrepancy between NPRS score and % subjective improvement after SIJ injection warrants further investigation into how to best document patient perception of success after injection.

### Limitations

There are several limitations of the study. Due to its retrospective nature, the follow up time for return to clinic ranged widely. Given the wide range of follow up times, some patients may have had their follow-up clinic visit after the effect of their injection had worn off, leading to their reported % improvement to be falsely lower than if recorded earlier, or what could have been seen when the steroid effect was present. This study also focused on patients’ first SIJ injection and so long-term outcomes for patients who receive multiple SIJ injections were not addressed. The responder rate was 40%, which is consistent with other retrospective studies that have reported the % of people who improve after a single injection. Hawkins and Schofferman [3] noted that in a cohort of 155 patients, there were 120 positive responders to SIJ injection (77%), and of those positive responders, one-third of patients improved after one injection, whereas two-thirds required multiple injections. Additionally, due to the retrospective nature of this study, it is likely that many of the patients who were lost to follow-up, and therefore excluded, were actually those who benefitted greatly from the injections, as patients for whom the injections gave complete pain relief are less likely to return to clinic for follow-up. In contrast, patients whose injection did not work or worked ephemerally are more likely to return to clinic for re-evaluation and to discuss additional treatment options. Additionally, our study focused on 10 specific variables for investigation as patient-specific predictors. While the predictors were selected based on their known association with low back pain, which can be in part due to SIJ pain, there are many other possible predictors that have been suggested to affect low back pain, such as alcohol intake [24], and these variables may also affect response to SIJ injection. Further studies investigating these predictors in a larger cohort are warranted.

## 5. Conclusions

A retrospective chart review at an academic multi-specialty pain center revealed that a history of anxiety and depression and increased age significantly decreased responsiveness to initial SIJ injection. Additionally, the associated NPRS score change for patients who were classified as responders in our study was less than the minimally clinically significant value of a 2-point differential, suggesting patient perception of success after SIJ injection may not be accurately represented through changes in pain scores. A prospective study evaluating these patient-specific predictors with follow-up at set time periods is warranted.

## Figures and Tables

**Figure 1 ijerph-19-15519-f001:**
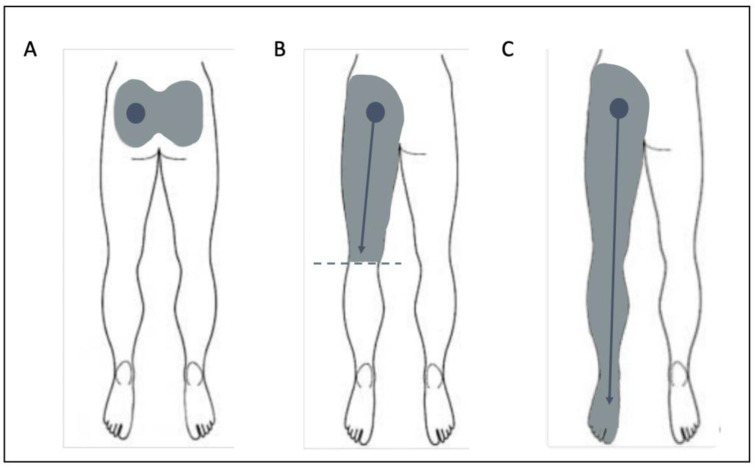
Pain Location Patterns. (**A**): lower back/buttocks only, (**B**): lower back/buttocks with pain radiating down to knees only, (**C**): lower back/buttocks with pain radiating past knees. Pain Location pattern D (not pictured) is lower back/buttocks with upper body pain. Diagrams adapted from Jung et al. [7] with permission.

**Figure 2 ijerph-19-15519-f002:**
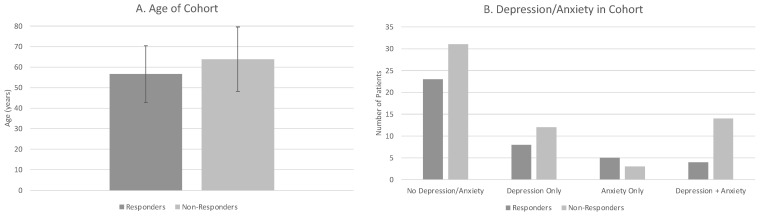
Significant Predictors of Response to Sacroiliac Joint Injections. (**A**) Average age of responders and non-responders. (**B**) Frequency comparisons of depression/anxiety status of responders and non-responders.

**Table 1 ijerph-19-15519-t001:** Patient demographic characteristics across responder categories.

		Responders (R) (n = 40)	Non-Responders (NR) (n = 60)	*p*
Age, mean ± SD years		57 ± 14	64 ± 16	0.019
Gender,	n(%)			
	Male	10 (25.0)	17 (28.3)	
	Female	30 (75.0)	43 (71.7)	
BMI, mean ± SD		28 ± 5.2	28 ± 6.7	0.880
% improvement, mean ± SD		74 ± 16%	24 ± 28%	<0.001
Duration of Improvement, mean ± SD, months		4.1 ± 3.0	1.1 ± 1.3	<0.001
Change in NPRS, mean ± SD points		−1.1 ± 2.4	−0.25 ± 1.5	0.034
Occupation, n(%)				
Manual Labor		4 (10%)	5 (8.3%)	
Sedentary Job		10 (25%)	13 (22%)	
Homemaker		1 (3%)	5 (8.3%)	
Unemployed/On Disability		13 (33%)	11 (18%)	
Retired		7 (18%)	15 (25%)	
Unknown		5 (13%)	10 (17%)	

**Table 2 ijerph-19-15519-t002:** Spine specific risk factors investigated as predictors.

	Responders (R) (n = 40)	Non-Responders (NR) (n = 60)	*p*	Odds Ratio (95% Confidence Interval)
H/o Lumbar Spine Surgery, n(%)			0.555	0.698 (0.211–2.302)
Yes	6 (15.0)	13 (21.7)		
No	34 (85.0)	47 (78.3)		
Pain Referral Pattern, n(%)				
A (reference)	14 (35.0)	16 (26.7)	0.349	
B	11 (27.5)	22 (36.7)	0.189	0.450 (0.136–1.483)
C	5 (12.5)	13 (21.7)	0.261	0.443 (0.107–1.831)
D	10 (25.0)	9 (15.0)	0.893	1.094 (0.296–4.039)
H/o Depression/Anxiety, n(%)				
No Depression or Anxiety (reference)	23 (57.5)	31 (51.7)	0.131	
Depression Only	8 (20.0)	12 (20.0)	0.425	0.610 (0.181–2.055)
Anxiety Only	5 (12.5)	3 (6.67)	0.424	1.996 (0.367–10.860)
Depression + Anxiety *	4 (10.0)	14 (23.3)	0.043 *	0.233 (0.057–0.954) *
Opioid Risk Tool Scores, mean ± SD	3.3 ± 5.1	2.7 ± 3.4	0.920	0.993 (0.875–1.128)

History of depression and anxiety significantly reduced the odds of responding to injections. History of lumbar spine surgery, pain referral pattern, and opioid risk tool scores were not significantly associated with response to sacroiliac joint injections. * = *p* < 0.05.

**Table 3 ijerph-19-15519-t003:** Demographic variables and comorbidities investigated as predictors.

	Responders (R) (n = 40)	Non-Responders (NR) (n = 60)	*p*	Odds Ratio (95% Confidence Interval)
Age, mean ± SD years *	57 ± 14	64 ± 16	0.006 *	0.946 (0.910–0.984) *
Gender, n(%)			0.780	0.858 (0.293–2.511)
Male	10 (25.0)	17 (28.3)		
Female	30 (75.0)	43 (71.7)		
BMI, mean ± SD	28.1 ± 5.27	28.3 ± 6.75	0.489	0.972 (0.897–1.054)
Smoking Status, n(%)				
Never Smoked (reference)	16 (40.0)	26 (43.3)	0.602	
Former Smoker	18 (45.0)	25 (41.7)	0.314	1.722 (0.598–4.955)
Currently Smokes	6 (15.0)	9 (15.0)	0.702	1.367 (0.275–6.786)
H/o of Diabetes, n(%)			0.329	0.503 (0.126–2.001)
Yes	5 (12.5)	10 (16.7)		
No	35 (87.5)	50 (83.3)		
H/o of Hypertension, n(%)			0.144	2.222
Yes	21 (40.0)	21 (35.0)		
No	24 (60.0)	39 (65.0)		

Increased age significantly reduces the odds of responding to injections. Gender, BMI, smoking status, history of diabetes, or history of hypertension were not significantly associated with response to sacroiliac joint injections. * = *p* < 0.05.

## Data Availability

The data presented in this study are available on request from the corresponding author. The data are not publicly available as they can contain patient identifying data.

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
