# Peer review of "A Retrospective Study on Patient-Specific Predictors for Non-Response to Sacroiliac Joint Injections"

_ijerph, 2022, doi:10.3390/ijerph192315519_

Round 1

Reviewer 1 Report

The paper sounds very interestingly, since it declares it wants to investigate patients-related predictors of response to sacro-iliac  joint injections. Moreover, it enrolled 100 patients, a quite large number of patients.

However, along the entire results and discussion  session the authors focused on those factors that are associated with reduced response to the block, such as anxiety and depression. 

These are well known factors that are associated with reduced outcome with all invasive techniques, and therefore its is not  a surprise to found them also in this paper. No data are shown on positive predictive factors, as the title suggested.  Have the authors tried to investigate which provocative exam could better help in diagnosing SIJ pain, or which combination of signs and symptoms? No data are available on clinical data that really can help in discriminating SIJ pain, and therefore, the efficacy of intra-articular injection.

Moreover, as the authors suggested, maybe a single NPRS evaluation could not be sufficient to identify responders, since other aspects should be investigated, such as disability, quality of life, activity of daily living, and so on. 

Finally, I do not understand figure 1: what is the difference between panel 2 and 3?

Reviewer 2 Report

It is a very well written manuscript and very interesting results. I looked into the reference number 15 because I found figure 1 lacking more information. I could not find this figure in reference number 15 but there were other figures with similar information. I would like to see a better figure to describe for the readers these pain location patterns. Table 2 Depression/anxiety is a little bit confusing. I found it difficult to see the results which were presented about those particular results.

Reviewer 3 Report

Dear authors,
I am, on the whole, in favour of your manuscript ´A Retrospective Study on Patient-Specific Predictors for Responsiveness to Sacroiliac Joint Injections´, though I have some questions especially regarding your methods and selection of patients.
These are:
1. Who are the patients? Are they referred to your clinic from primary care or do and what treatment had they had on beforehand? How long did they have pain in that particular region? What professions did they have? Were they on sick-leave? How were there work load at home? How much of house-hold duties could they do? How about the women - did they have post-partum girdle pain?
2. Were the included in the study consecutive patients?
3. I gather this study was not designed from the beginning as a research study. Is your study rather an observation study, or quality survey of your clinical work in ´real life´?    
4. Line 97. Was NPRS studied before and after or only once, i.e. at the first follow-up?
5. Line 103. What has smoking to do with sacro-iliac joint pain? You say nothing about alcohol intake for instance.    
6. Line 104. I am not sure of the relevance of diabetes and hypertension in this sacro-iliac joint pain. References 12 and 13 concerns low back pain in general.
7. Lines 91-105 do not mention anything about disability, work-ability, post-partum pain, house-hold duties and physical activity. These things would be valuable to know for the readers before and after of your intervention.
8. Table 1. Again - NPRS - how and when was it measured?
9. Table 2. Needs to be improved in design to make it easier to read.

Good luck!

Round 2

Reviewer 1 Report

The  paper is improved and describes more clearly its ppopulation. 

Only one question: did the authors investigate the imaging of these patients? Were there signs of SIJ arthritis, spondyloarthropathies, ligamentous and muscular injuries, and enthesopathy? Did all patients undergo strumental examination?

Was the first follow up programmed? After how many days? and therefore, the reduction of NRS referred to this follow up? Moreover, did this reduction last for all 4 months? Did the authors noted any change of NRS over time?

Were the following recording  made by telephone or by other programmed visits? Did the patients reported other imrovement (for example in daily activity)?

Please specify.
